environmental engineering

U(VI), bioadsorption, *Bacillus megaterium*, sodium alginate

**Author for correspondence:**
Dianxin Li
e-mail: dianxin-li@foxmail.com

# U(VI) adsorption in water by sodium alginate modified *Bacillus megaterium*

Dianxin Li, Yiqing Yang, Peng Zhang, Jiangang Liu, Tao Li and Junwei Yang

School of Mines and Civil Engineering, Liupanshui Normal University, 288 Minghu Road, 553004 Liupanshui, People's Republic of China

DL, 0000-0002-4531-0112

The surface of *Bacillus megaterium* was modified by coating sodium alginate. The modified *B. megaterium* before and after adsorption were characterized by SEM, FTIR and XPS. The effects of pH, reaction time, initial U(VI) concentration and adsorbent dosage on the adsorption of U(VI) by the modified *B. megaterium* were studied by batch adsorption experiments. The adsorption process was studied by pseudo-first-order kinetics and pseudo-second-order kinetic models, Langmuir and Freundlich isotherms. The results showed that the maximum adsorption capacity of U(VI) was 74.61 mg g$^{-1}$ under the conditions of pH 5.0, adsorbent 0.2 g l$^{-1}$, 30°C and initial U(VI) concentration of 15 mg l$^{-1}$. The adsorption process accords with pseudo-first-order kinetics and Langmuir isotherm. The adsorption capacity of U(VI) by the modified *B. megaterium* was still higher than 80% after five times of desorption and reuse experiments. In conclusion, the sodium alginate modified *B. megaterium* was an ideal material for U(VI) biosorption.

## 1. Introduction

With uranium mining, hydrometallurgy, processing, enrichment and other nuclear industrial activities, a large amount of uranium wastewater has been produced [1]. If the uranium wastewater is not treated properly, it will move with groundwater or surface water, which will seriously threaten human health [2]. In recent years, the main remediation methods of uranium wastewater include chemical precipitation, biological remediation (including phytoremediation and microbial remediation), adsorption and ion exchange. Currently, chemical precipitation method is a usually used method at present, but when the dosage of chemical reagent is excessive, it will cause secondary pollution [3]. Uranium bioremediation in groundwater by microorganism shows that once the electron donors are stopped in the injection well, the reduced U(IV) will be oxidized to U(VI) quickly. It

cannot achieve the long-term remediation [4]. Phytoremediation is an effective method to treat uranium contamination in topsoil [5]. Adsorption method has the advantages of a wide selection of adsorbents, high adsorption efficiency, high recovery and good economic effect, which has been widely studied by domestic and foreign scholars [6]. Biosorption is not only an adsorption method, but also a bioremediation method, which has both advantages [7]. It uses plants, microorganisms, animal bones and other biological materials as adsorption materials, through reasonable modification, to achieve the purpose of adsorption of heavy metals and radioactive metals [8]. This method has the characteristics of wide sources of materials, low cost and no secondary pollution, so it should be widely studied.

*Bacillus* is a common water treatment microorganism, which has the advantages of wide distribution, fast growth rate and many metabolites [9]. It is widely used in the remediation of heavy metal pollution. In this experiment, *Bacillus megaterium*, which has the advantages mentioned above, was used to synthesize biosorbent materials [10]. Sodium alginate is usually used to encapsulate the cells and metabolites of microorganisms to prepare biosorbent materials [11]. Recently, Jiang *et al.* [12] have shown that sodium alginate can also improve the adsorption capacity of U(VI).

In this study, *B. megaterium* was coated with sodium alginate and modified to form a biosorbent material. This material not only integrates the advantages of sodium alginate and *B. megaterium*, but also uses sodium alginate to modify the surface of *B. megaterium*. Due to *B. megaterium* being a microorganism existing in the environment, it causes little secondary pollution to the environment. So it is a kind of low price, little environmental harm and good adsorption effect of U(VI) biosorption material. The specific research works were as follows: (i) batch adsorption experiments were carried out to analyse the effects of pH (the strength of acids and alkalis), reaction time, initial U(VI) concentration and adsorbent dosage on adsorption; (ii) the modified *B. megaterium* before and after adsorption were characterized by SEM, FTIR and XPS; (iii) the adsorption process was studied by combining the pseudo-first-order kinetic model, pseudo-second-order kinetic model, Langmuir and Freundlich isotherms; and (iv) desorption reuse experiment was used to explore the reuse times of modified *B. megaterium*.

# 2. Materials and methods

## 2.1. Materials

Experimental reagents: sodium alginate (Shanghai Titan Technology Co., Ltd, China), $U_3O_8$ (Xi'an Dingtian Chemical Co., Ltd, China), hydrochloric acid and nitric acid (Guiyang Yida Chemical Reagent Co., Ltd, China), LB medium (Qingdao Hope Biotechnology Co., Ltd), potassium dihydrogen phosphate, potassium hydrogen phosphate and calcium chloride (Tianjin Kaitong Chemical Reagent Co., Ltd, China, analytical grade).

Strain: *B. megaterium* (China General Microbiological Culture Collection Center, No.: 1.10466).

## 2.2. Synthesis of *Bacillus megaterium*

Extraction of *B. megaterium* biomass. *B. megaterium* was inoculated in LB medium sterilized at 121°C for 30 min and cultured at 150 r.p.m. and 30°C for 24 h. After cultured for 24 h, *B. megaterium* biomass was centrifuged at 4000 r.p.m. for 10 min and washed with phosphate buffer for three times to obtain *B. megaterium* biomass [13].

Surface modification of *B. megaterium*. Two grams of sodium alginate was added into 98 ml deionized water (w/v, 2%), heating the water bath and stirred continuously until sodium alginate was completely dissolved [14]. Five grams of *B. megaterium* biomass (wet weight) was added into the sodium alginate solution and dissolved by ultrasonic wave to obtain *B. megaterium* and sodium alginate mixture evenly [15]. The mixture was added to the calcium chloride solution (w/v, 2%) drop by drop with a 10 ml syringe with the needle removed to obtain *B. megaterium* pellets (BM-SA) wrapped with sodium alginate. The mixture was washed with distilled water three times and dried in vacuum at 60°C for 12 h.

## 2.3. Characterization

Surface modified *B. megaterium* (BM-SA) was used as an adsorbent for U(VI) in solution. The samples before and after biosorption were analysed by scanning electron microscopy (SEM, FEI

QUANTA400FEG, USA), Fourier transform infrared spectroscopy (FTIR, VERTEX70, Brooke) and X-ray photoelectron spectroscopy (XPS, Thermo ESALAB 250xi, USA) [16]. The acquisition range of FTIR spectrum was 400–4000 cm$^{-1}$. The scanning range of XPS was 0–1320 eV, and the data were analysed by XPSPEAK4.1 software.

## 2.4. Batch adsorption experiments

The effects of pH, reaction time, initial U(VI) concentration and adsorbent dosage on adsorption of U(VI) by BM-SA were studied by batch adsorption experiments. Forty millilitres of uranium standard solution (1–15 mg l$^{-1}$) was taken into a 50 ml centrifuge tube, and the pH value of the solution was adjusted with 1 mol l$^{-1}$ HCl or NaOH solution. The centrifuge tube was placed in a shaker and the adsorption experiment was carried out at 30°C and 150 r.p.m. [17].

U(VI) adsorption capacity (mg g$^{-1}$) and adsorption rate (%) were calculated by the formula (2.1) and formula (2.2), respectively [18]. The concentration of U(VI) was determined by ArsenazoIII spectroscopic method [19].

$$q = \frac{(C_0 - C_e) \times V}{W} \tag{2.1}$$

and

$$R = \frac{C_0 - C_e}{C_0} \times 100\%, \tag{2.2}$$

where $C_0$ and $C_e$ were the initial and equilibrium concentrations of U(VI), respectively. $V$ was the volume of U(VI) in solution ($L$), and $W$ was the weight of adsorbent ($g$).

## 2.5. Desorption and regeneration experiments

The reuse time of adsorbent was studied, and five times of desorption and reuse experiments were carried out. Eight milligrams of adsorbed biomaterial was placed in 40 ml 1 mol l$^{-1}$ HCl, reacted for 12 h at 30°C and 150 r.p.m. in a shaking table. After centrifugation, the supernatant was taken to measure the concentration of U(VI), to calculate the desorption percentage of U(VI) adsorbed by BM-SA. The adsorbent was obtained by filtering the solution, washed with deionized water three times and dried in a vacuum drying oven at 60°C for 12 h [20]. The newly prepared U(VI) solution (15 mg l$^{-1}$, pH 6.0) was used for the adsorption experiment.

# 3. Results and discussion

## 3.1. Characteristics of BM-SA

The surface morphology of BM-SA was characterized by SEM. According to figure 1a, BM-SA was approximately ellipsoidal in structure. This was due to the chemical reaction of sodium alginate coated with *B. megaterium* and calcium chloride solution [21]. The surface structure of BM-SA was uneven, indicating that it had a large specific surface area, which was conducive to its adsorption of U(VI) in solution (figure 1b) [22]. After the adsorption of U(VI) by BM-SA, a stacked irregular spherical structure was formed on the surface of BM-SA (figure 1c), indicating that U(VI) could be well adsorbed by BM-SA.

The FTIR spectra of U(VI) before and after adsorption are shown in figure 2. It could be seen from figure 2a, there were N-H, C-H, C≡C stretching vibration characteristic peaks at wave numbers of 3268.19, 2920.03, 2162.76 cm$^{-1}$ [23], N-H bending vibration characteristic peaks at wave number 1595.57 cm$^{-1}$, C-N stretching vibration (or C-H bending vibration) characteristic peaks at wave number 1417.46 cm$^{-1}$, C-O stretching vibration (or C-H bending vibration) characteristic peaks appeared at 1078.58 and 1026.56 cm$^{-1}$ and the characteristic peaks of C-H bending vibration appeared at wave numbers of 882.79 and 816.27 cm$^{-1}$ [24]. After adsorption, the above characteristic peaks were shifted to a certain extent. Among them, the characteristic peaks of C-H stretching vibration and C-H bending vibration at wave numbers 2920.03 and 882.79 cm$^{-1}$ disappeared, indicating that the adsorption of U (VI) may be most related to C-H [25].

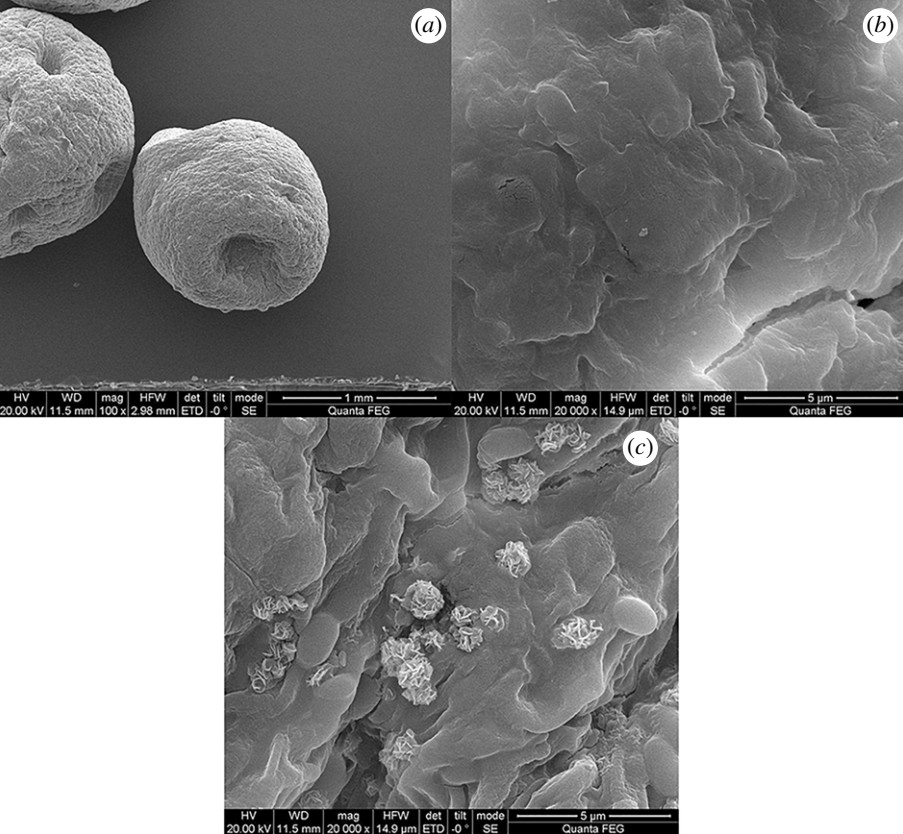

**Figure 1.** SEM images of BM-SA before adsorption (*a,b*) and after adsorption (*c*).

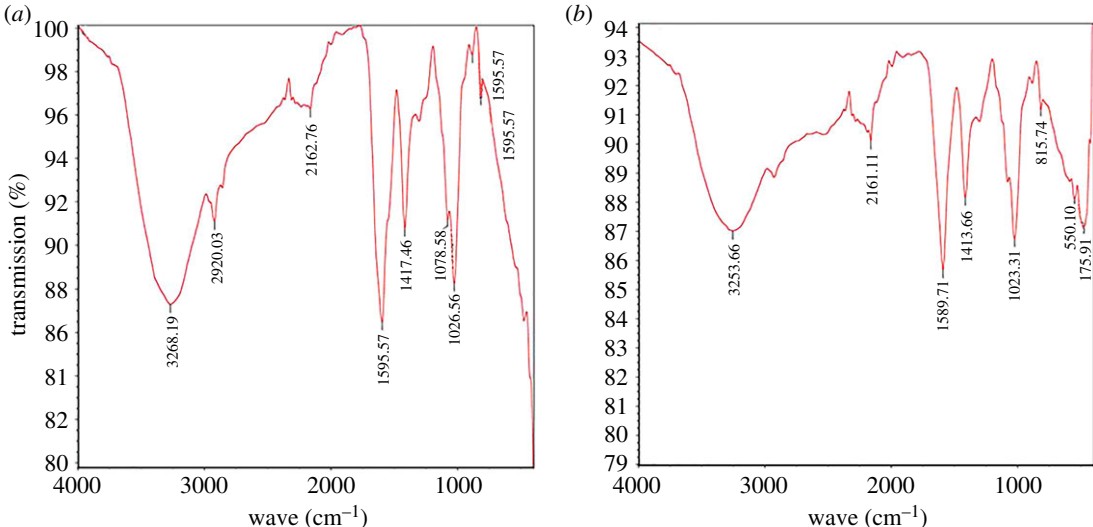

**Figure 2.** FTIR spectra of BM-SA before (*a*) and (*b*) after adsorption.

## 3.2. Effect of pH

pH was one of the important parameters that affect the adsorption of U(VI) by adsorbent [26]. In this experiment, 8 mg adsorbent was used to study the adsorption effect of BM-SA on U(VI) under the conditions of initial U(VI) concentration of 15 mg l$^{-1}$, 6 h, and pH adjusted to 3, 4, 5, 6, 7, 8 and 9, respectively (figure 3*a*). Visual MINTEQ 3.1 software was used to simulate the morphology of U (VI) in the solution at the concentration of 15 mg l$^{-1}$ and pH 3–9 (figure 3*b*) [27]. It could be seen from figure 3*a* that when pH < 5, the adsorption capacity of BM-SA for U(VI) increased with the increasing

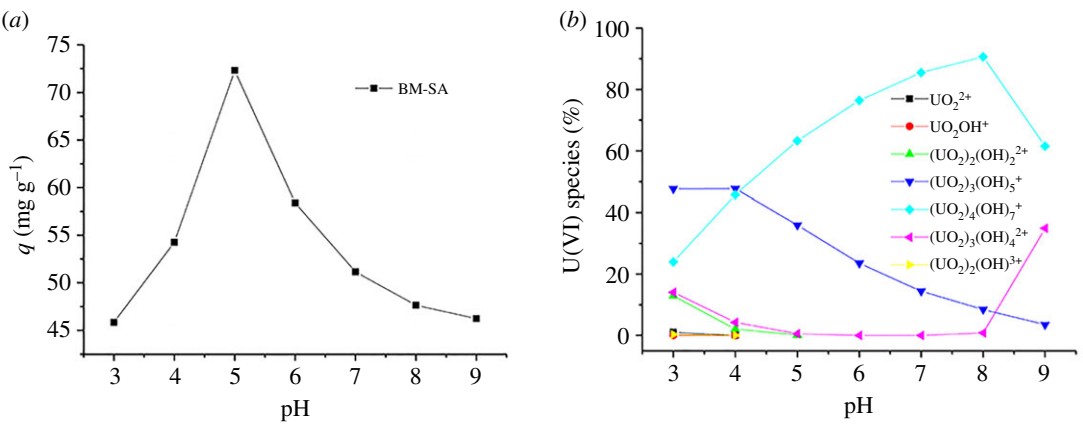

**Figure 3.** The effect of pH on the adsorption of U(VI) by BM-SA (*a*) and the form of U(VI) at different pH (*b*) ([U(VI)] = 15 mg l$^{-1}$, contact time: 6 h, temperature: 30°C, BM-SA: m/v = 0.2 mg l$^{-1}$).

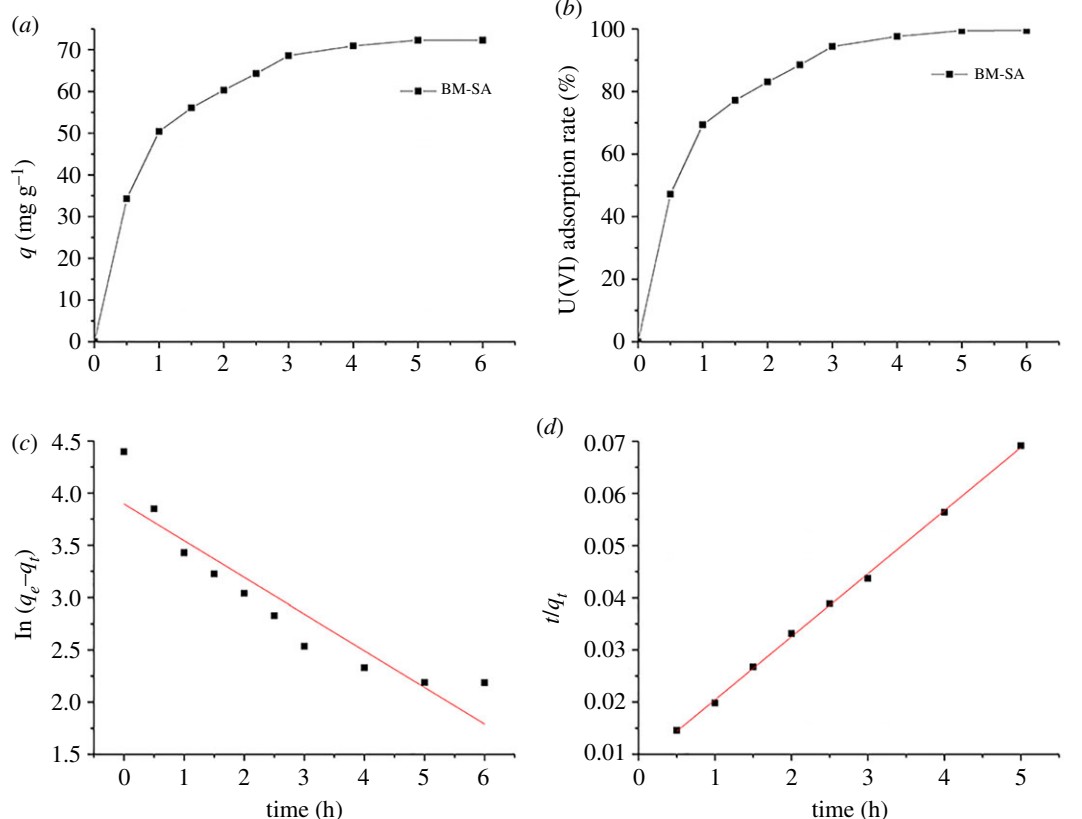

**Figure 4.** The effect of time on the adsorption capacity (*a*) and adsorption rate (*b*) of BM-SA for U(VI) adsorption and the pseudo-first-order kinetics (*c*) and pseudo-second-order kinetics (*d*) by linear fitting ([U(VI)] = 15 mg l$^{-1}$, pH: 5.0, temperature: 30°C, BM-SA: m/v = 0.2 mg l$^{-1}$).

of pH. When pH > 5, the adsorption rate of U(VI) by BM-SA decreased with the increasing of pH. Therefore, the most suitable pH for U(VI) adsorption by BM-SA was 5. It can be seen from figure 3*a* that when pH was 5, the main forms of U(VI) in the solution were $(UO_2)_4(OH)_7^+$ and $(UO_2)_3(OH)_5^+$, indicating that BM-SA has good adsorption effect on these two forms of U(VI).

### 3.3. Effect of reaction time and kinetics

The effect of reaction time on adsorption of U(VI) by BM-SA was studied at pH 5 using 8 mg adsorbent. It could be seen from figure 4*a,b* that the adsorption rate of U(VI) by BM-SA was faster within 0.5 h; the

**Table 1.** Kinetic model parameters of adsorption of U(VI) by BM-SA.

| pseudo-first-order kinetics | | | pseudo-second-order kinetics | | |
|---|---|---|---|---|---|
| $k_1$ (h$^{-1}$) | $q_e$ (mg g$^{-1}$) | $R^2$ | $k_2$ (mg g$^{-1}$ h$^{-1}$) | $q_e$ (mg g$^{-1}$) | $R^2$ |
| 1.16 | 70.71 | 0.85 | 0.02 | 81.82 | 0.99 |

adsorption rate decreased in 0.5–5 h; the adsorption almost reached equilibrium after 5 h [28]. The adsorption capacity was 72.29 mg g$^{-1}$ and the adsorption rate was 99.4%. The adsorption rate of U(VI) was faster in 0.5 h because the adsorbent showed more adsorption sites and was easy to be adsorbed. With the process of adsorption, the adsorption sites on the adsorbent surface gradually decreased, resulting in the decrease of adsorption rate [29].

In order to further study the interaction mechanism between BM-SA and U(VI), the pseudo-first-order kinetic model (equation (3.1)) and the pseudo-second-order kinetic model (equation (3.2)) were used for linear fitting [30]. In the kinetic adsorption experiment, 8 mg SA-GO was added to a 40 ml solution of 15 mg l$^{-1}$ at pH 5, 30°C [31].

$$\ln(q_e - q_t) = \ln q_e - k_1 t \tag{3.1}$$

and

$$\frac{t}{q_t} = \frac{1}{k_2 q_e^2} + \left(\frac{1}{q_e}\right) t, \tag{3.2}$$

where $t$ was the reaction time (h), $q_e$ and $q_t$ represent the equilibrium adsorption capacity and the adsorption capacity at $t$ time (mg g$^{-1}$), respectively; $k_1$ (h$^{-1}$) and $k_2$ (mg g$^{-1}$ h$^{-1}$) were the constants of pseudo-first-order kinetic model and pseudo-second-order kinetic model, respectively, which obtained by the linear fitting.

Compared with figure 4c and figure 4d, the adsorption of U(VI) by BM-SA was more consistent with the pseudo-second-order kinetic model. The $R^2$ of pseudo-second-order kinetics was 0.99, which was higher than that of pseudo-first-order kinetics (0.85) (table 1) [32]. Under certain conditions, $q_e$ was a constant, and $q_t$ increased gradually with time and approached $q_e$. So $\ln(q_e - q_t)$ was a decreasing function (figure 4c). The equilibrium adsorption capacity of the pseudo-first-order kinetic model and the pseudo-second-order kinetic model were 70.71 and 81.82 mg g$^{-1}$, respectively. Some studies showed that the maximum adsorption capacity of the adsorption capacity was between the pseudo-first-order and the pseudo-second-order kinetic model [33]. Further analysis was needed to determine the maximum adsorption capacity.

## 3.4. Effect of initial U(VI) concentration and adsorption isotherm

The initial U(VI) concentration had a certain influence on U(VI) adsorption [31]. The effects of initial U(VI) concentrations of 1 ,3, 5, 10 and 15 mg l$^{-1}$ on U(VI) adsorption by BM-SA were studied. The conditions for obtaining the adsorption capacity $q$ were as follows: adsorbent 8 mg, adsorption time 6 h, pH 5. It could be seen from figure 5a that with the increasing of initial U(VI) concentration, the adsorption capacity of BM-SA for U(VI) also increased and tends to balance gradually. The results showed that BM-SA had better adsorption effect on U(VI) when the initial U(VI) concentration was lower than 15 mg l$^{-1}$.

In order to study the concentration relationship between the solid phase and liquid phase, Langmuir (formula (3.3)) and Freundlich (formula (3.4)) isotherm models were used to fit the concentration relationship of adsorbate between the solid phase and liquid phase [34].

$$\frac{c_e}{q_e} = \frac{1}{k_L q_m} + \frac{c_e}{q_m} \tag{3.3}$$

and

$$\ln q_e = \frac{1}{n_f} \ln c_e + \ln k_f, \tag{3.4}$$

where $c_e$ was the equilibrium concentration (mg l$^{-1}$); $q_e$ and $q_m$ represented the equilibrium adsorption capacity and maximum adsorption capacity (mg g$^{-1}$); $k_L$ (l mg$^{-1}$) was the Langmuir isotherm

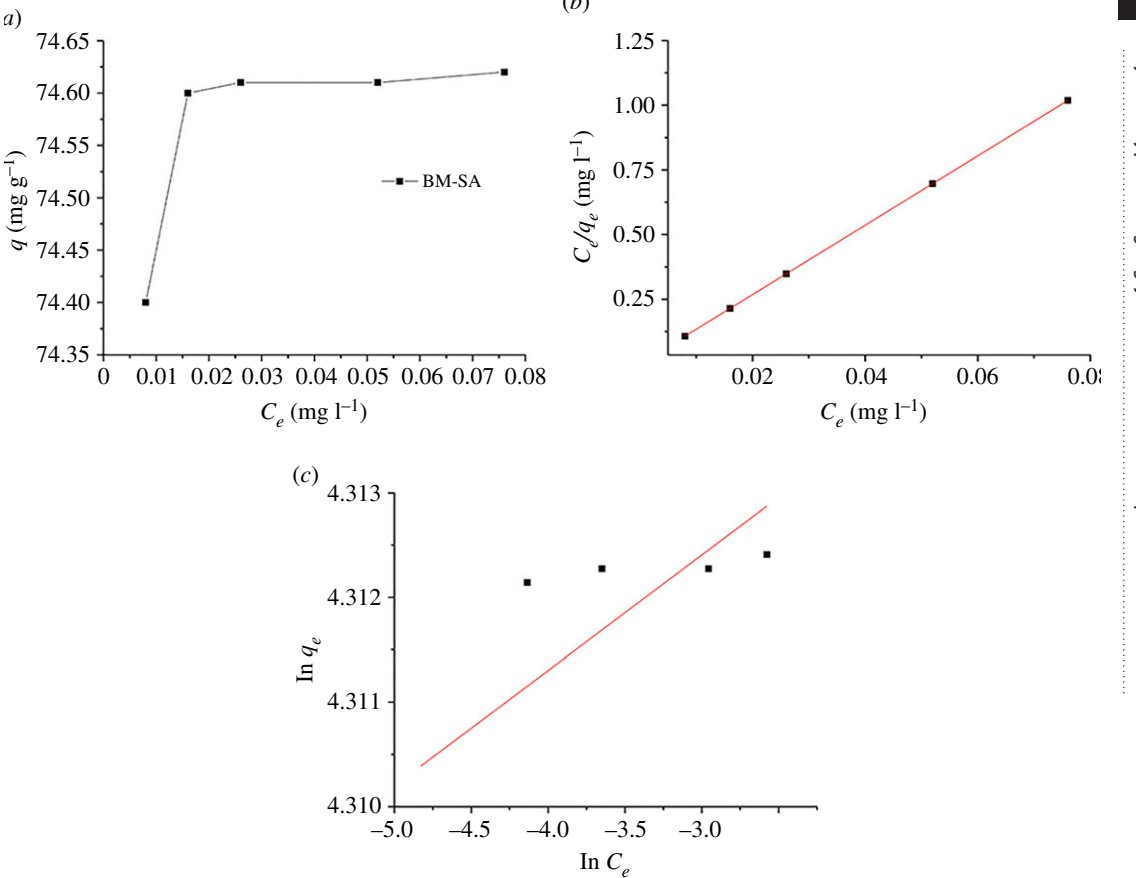

**Figure 5.** Adsorption isotherms (*a*), and linear fitting of isotherms model Langmuir (*b*) and Freundlich (*c*) of U(VI) adsorption by BM-SA (contact time: 6 h, pH: 5.0, temperature: 30°C, BM-SA: m/v = 0.2 mg l$^{-1}$).

**Table 2.** Isotherm parameters of U(VI) adsorption by BM-SA(Keep 2 decimal places).

| Langmuir isotherm | | | Freundlich isotherm | | |
|---|---|---|---|---|---|
| $q_m$ (mg g$^{-1}$) | $K_L$ (L mg$^{-1}$) | $R^2$ | $n_f$ | $K_f$ [(mg g$^{-1}$)/(L/mg)$^{1/n}$] | $R^2$ |
| 74.61 | 3.95 | 0.99 | 74.87 | 907.91 | 0.49 |

constant, $k_f$ ((mg g$^{-1}$)/(l mg$^{-1}$)$^{1/n}$) and $n_f$ were the Freundlich isotherm constants, respectively, which were obtained by liner fitting.

It could be seen from figure 5*b* and figure 5*c* that the adsorption process conformed to the Langmuir isotherm model, which $R^2$ was 0.99; while the $R^2$ of Freundlich was only 0.49 [35]. The maximum adsorption capacity $q_m$ was 74.61 mg g$^{-1}$ (table 2), which was between the fitting results of pseudo-first-order kinetic model and pseudo-second-order kinetic model, and was close to the experimental equilibrium adsorption capacity $q_m$ (72.29 mg g$^{-1}$). Compared with the previous studies, BM-SA has better adsorption capacity for U(VI) (table 3).

## 3.5. Effect of adsorbent dosage

The amount of adsorbent had a significant effect on U(VI) adsorption [36]. In this experiment, 2, 4, 6, 8 and 10 mg BM-SA were used to react for 6 h at pH 5, 15 mg l$^{-1}$ U(VI). It could be seen from figure 6 that when the dosage of BM-SA was less than 8 mg, the adsorption capacity of BM-SA on U(VI) increased with the increasing of adsorbent dosage. When the dosage of BM-SA was higher than 8 mg, the adsorption capacity of U(VI) was not increased by increasing the amount of adsorbent. Therefore, when the initial U(VI) concentration was 15 mg l$^{-1}$, the most suitable dosage of BM-SA was 0.2 g l$^{-1}$ (8 mg (BM-SA)/40 ml (solution)).

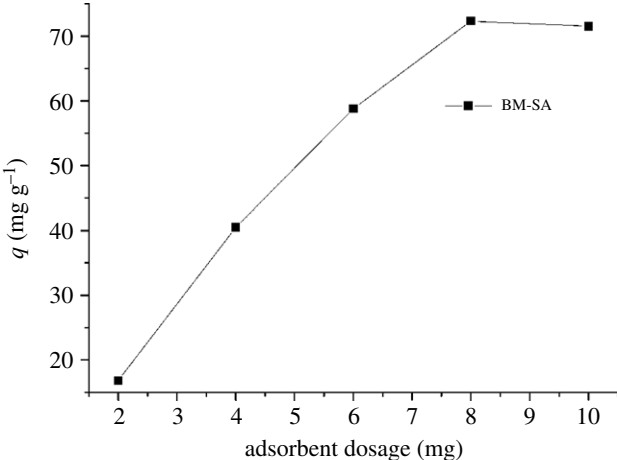

**Figure 6.** Effect of adsorbent dosage on adsorption of U(VI) by BM-SA ([U(VI)] = 15 mg l$^{-1}$, pH: 5.0, temperature: 30℃, contact time: 6 h).

**Table 3.** Comparison of maximum adsorption capacity of different adsorbents.

| adsorbent | adsorption conditions | $q_m$ (mg g$^{-1}$) | reference |
|---|---|---|---|
| *Bacillus* sp. | pH 3.0, 30℃ | 3.03 | [37] |
| *Pleurotus ostreatus* | pH 4.0 | 19.95 | [38] |
| modified *Spirulina platensis* | — | 73 | [39] |
| sugar beet pulp | — | 20.45 | [40] |
| *Solanum incanum* leaves | pH 4.0, 45℃ | 39.98 | [41] |
| *Bacillus megaterium* modified by sodium alginate | pH 5.0, 30℃ | 74.61 | this study |

## 3.6. Adsorption mechanism

The interaction mechanism between BM-SA and U(VI) was analysed by XPS. The full spectrum of U(VI) before and after adsorption and the high-resolution spectra of C1s, N1s, O1s and U4f were measured, respectively. It could be seen from the XPS spectrum that C1s, N1s and O1s peaks exist before U(VI) adsorption, and U4f peak appeared after adsorption (figure 7a) [42]. Before adsorption, the binding energies of C1s at 287.68, 285.98 and 284.28 correspond to C=O, H$_2$N-C=N-OH and C-(C,H), respectively (figure 7b); after adsorption, C=O was changed to C-O, and the binding energy was 286.68 (figure 7c) [25]. Before adsorption, the contents of C1s and O1s were 61.66% and 34.37%, respectively; after adsorption, they were 63.23% and 31.98% (table 4). In conclusion, U(VI) adsorption may be related to C=O.

The binding energy 399.48 (before adsorption) and 399.48 (after adsorption) correspond to the peaks of H$_2$N-C=N-OH (figure 8a), indicating that H$_2$N-C=N-OH had no effect on U(VI) adsorption. O1s spectrum showed that the peak of C-OH (binding energy 532.18) before adsorption was unchanged after adsorption (figure 8b), indicating that C-OH did not participate in U(VI) adsorption [25]. The images of U4f showed that there was no U(VI) on the surface of BM-SA before adsorption; after adsorption, peaks of U4f 5/2 and U4f 7/2 appeared at 392.38 and 381.48, respectively (figure 8c) [27]. The content of U4f also increased from 0 to 0.08% (table 4). These results indicated that BM-SA had a good adsorption effect on U(VI).

## 3.7. Desorption and reuse

The desorption and reuse of adsorbent was an important index to reflect its economic value [39]. In this experiment, 1 mol l$^{-1}$ HCl was used as the analytical agent, and the results are as shown in figure 9. In BM-SA, the resolution rate and reuse rate of U(VI) decreased with the increasing of resolution reuse cycles. After the fifth experiment, the resolution rate and reuse rate were 82.9% and 81.2%,

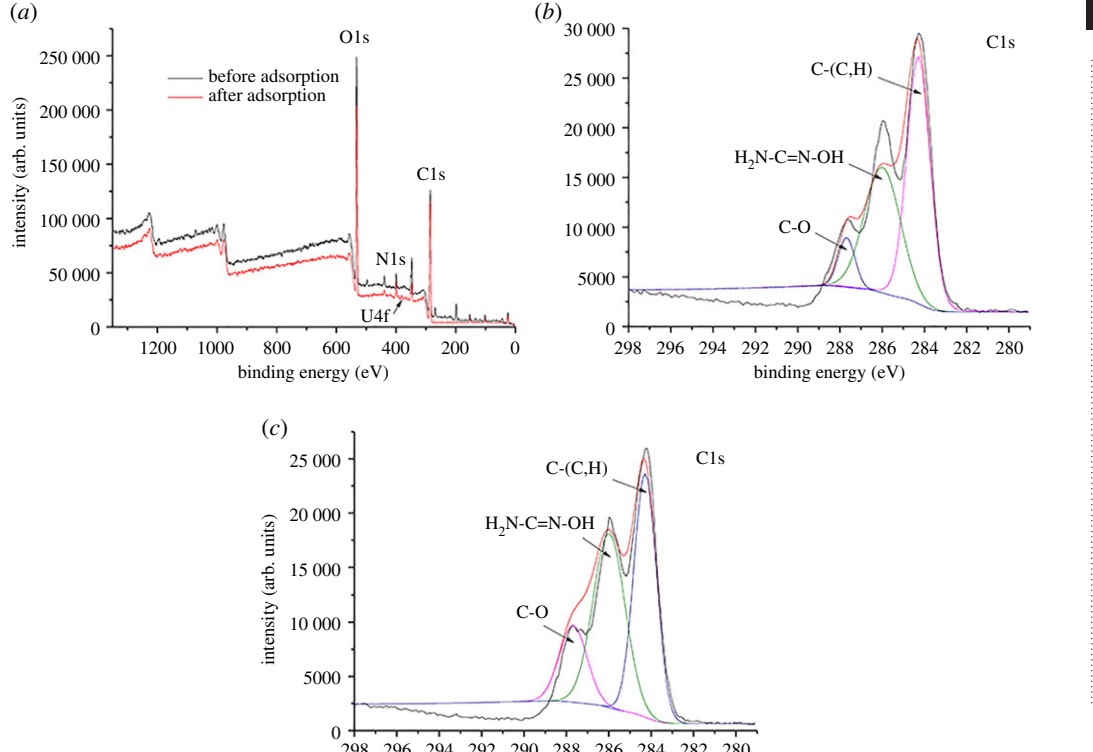

**Figure 7.** XPS scanning results of U(VI) adsorption on BM-SA. Full spectrum (*a*), high-resolution C1s spectra of before (*b*) and after adsorption (*c*).

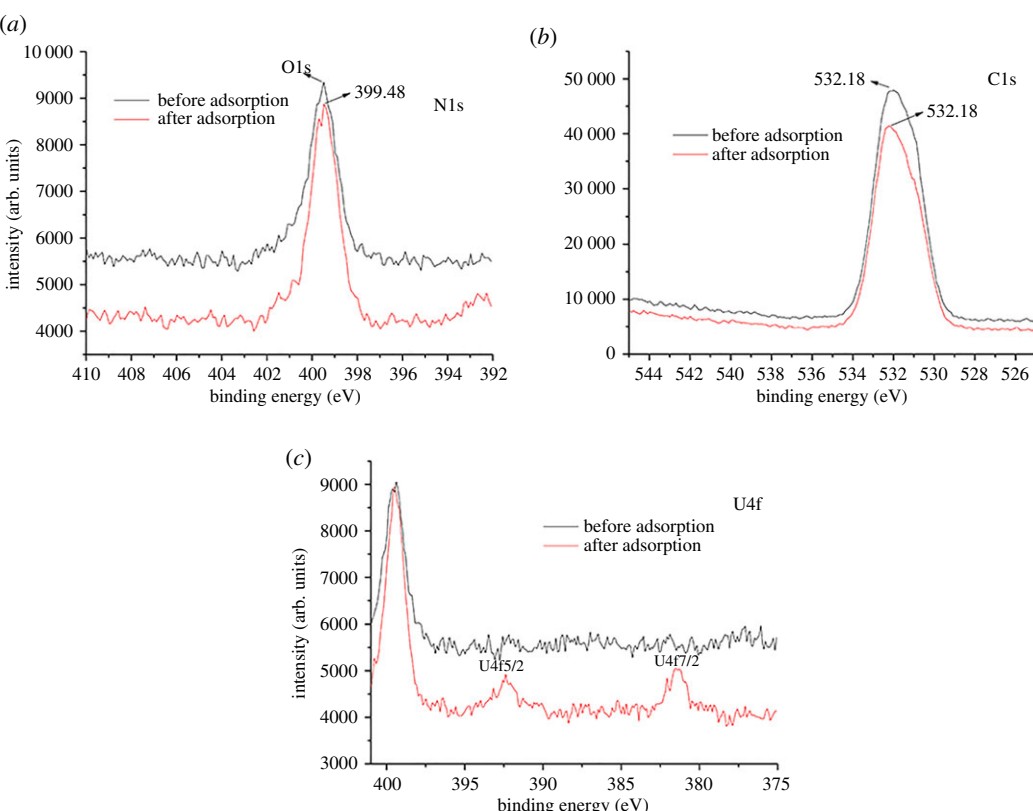

**Figure 8.** High-resolution spectra of N1s (*a*), O1s (*b*) and U4f (*c*) before and after U(VI) adsorption.

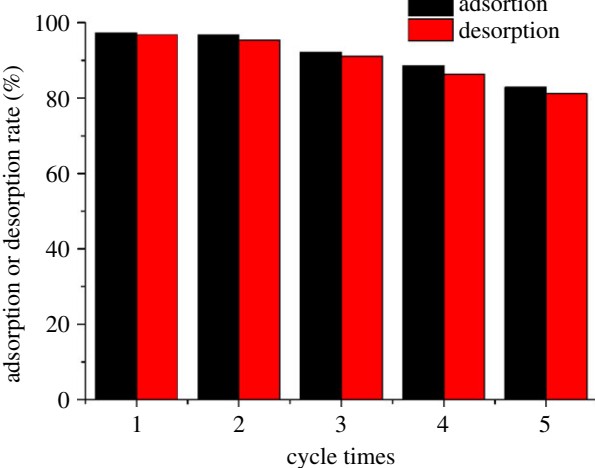

**Figure 9.** Resolution and reusability of BM-SA.

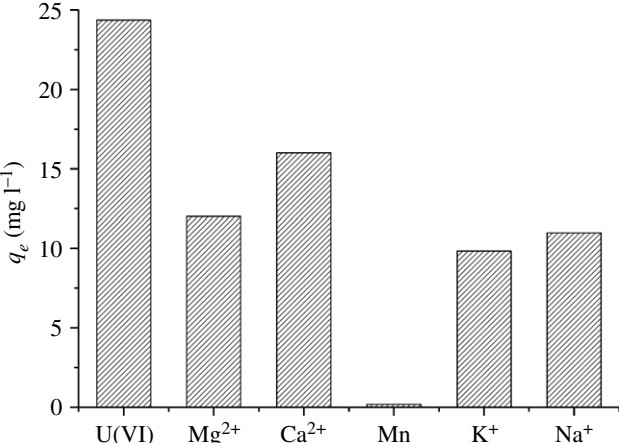

**Figure 10.** Selective adsorption of U(VI) by BM-SA (m/v = 0.2 g l$^{-1}$, 30°C, contact time 6 h).

**Table 4.** Elemental composition of BM-SA surface before and after U(VI) adsorption.

| name | C1s (%) | O1s (%) | N1s (%) | U4f (%) |
|---|---|---|---|---|
| before adsorption | 61.66 | 34.37 | 3.97 | 0 |
| after adsorption | 63.23 | 31.98 | 4.71 | 0.08 |

respectively, which were still at a high level. It proved that BM-SA had good regeneration function and had certain economic value.

## 3.8. Uranium removal from simulated uranium wastewater

The geochemical properties of the simulated uranium wastewater refer to our previous study [43]. The simulated uranium wastewater sample contains metal ions including Na$^+$ (171.4 mg l$^{-1}$), Ca$^{2+}$ (147.8 mg l$^{-1}$), K$^+$ (44.6 mg l$^{-1}$), Mg$^{2+}$ (16.8 mg l$^{-1}$), U(VI) (5.0 mg l$^{-1}$), Mn (0.2 mg l$^{-1}$), and its pH was 6.5. After the wastewater was treated by BM-SA, the removal efficiency of U(VI) was 97.5%, which was still at a high level in the absence of interfering ions [44].

## 3.9. Effect of other interfering ions

To evaluate the selectivity of BM-SA for U(VI) adsorption, selective adsorption experiments were conducted with the simulated uranium wastewater above. It could be seen from figure 10, BM-SA had

excellent binding selectivity for U(VI) adsorption than the other metal ions. The results suggested that BM-SA possessed higher selectivity for U(VI) adsorption over other interfering ions, and it had great potential for U(VI) enrichment from uranium removal in wastewater [44].

# 4. Conclusion

(1) In this study, the BM-SA biosorbent was prepared by modifying *B. megaterium* with sodium alginate.
(2) Batch adsorption experiments showed that the maximum adsorption capacity of BM-SA for U(VI) was 74.61 mg l$^{-1}$ at pH 5, 30°C and adsorbent dosage of 0.2 g l$^{-1}$, which compared with the single use of *Bacillus* sp., the adsorption performance of BM-SA was significantly improved. The adsorption may be the coordination of U(VI) with C-H and C=O on the surface of BM-SA. The adsorption accorded with the pseudo-second-order kinetic model, which indicated that the adsorption was mainly chemical adsorption. The adsorption process was more in line with the Langmuir isotherm, indicating that the adsorption was monolayer adsorption.
(3) The results showed that BM-SA had a good adsorption effect on U(VI) after five times of reuse.

In conclusion, BM-SA has the advantages of easy synthesis, high removal efficiency, economy and a certain prospect, which was a kind of U(VI) biosorption material for removing low concentration U(VI) in aqueous solution.

Data accessibility. We have uploaded our experimental dataset on Dryad; you can get it by the temporary link: https://doi.org/10.5061/dryad.fttdz08r9.

Authors' contributions. Conceptualization was done by P.Z.; methodology by D.L.; software by Y.Y.; validation by T.L.; formal analysis by J.Y.; investigation by J.L.; resources by P.Z.; data curation Y.Y.; original draft preparation was written by D.L.; review and editing were written by D.L.; visualization by T.L.; supervision by J.Y.; project administration by P.Z.; funding acquisition by D.L., J.L. and J.Y All authors have read and agreed to the published version of the manuscript.

Competing interests. We have no competing interests.

Funding. This research was funded by the Educational Department of Guizhou Province (qian jiao he KY[2019]138; qian jiao he KY[2018]029), Guizhou coal green development 2011 Collaborative Innovation Center (qian jiao he xie tong chuang xin zi[2016]02) and Liupanshui Normal University (LPSSYKYJJ201808).

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
