## [Peer Review File · Royal Society Open Science]

Review History

RSOS-202098.R0 (Original submission)

Review form: Reviewer 1

Is the manuscript scientifically sound in its present form?

Yes

Are the interpretations and conclusions justified by the results?

Yes

Is the language acceptable?

Yes

Do you have any ethical concerns with this paper?

No

Have you any concerns about statistical analyses in this paper?

No

Recommendation?

Accept with minor revision (please list in comments)

Comments to the Author(s)

In this manuscript, the authors modified *Bacillus megaterium* with sodium alginate, and characterized the material in detail. The sorption of U(VI) was studied under different experimental conditions and discussed the results in detail. The contents are important for the removal of U(VI) from aqueous solutions. After reading the manuscript, I think it can be accepted for publication after revision.

Special comments:

1. The study is been done only using pure Uranium solution, but no real or simulated solution was used as a case study. The author should use real or simulated solution uranium waste water.
2. In actual waste solution what are the interfering metal ions present and what is the effect of those metal ions on adsorption should also be included.
3. Page 2 Line 56-59: Please correct the sentence, "In recent years, the research on remediation of uranium wastewater pollution has emerged in an endless stream. Mainly contain chemical precipitation, bioremediation (including phytoremediation and microbial remediation), adsorption, and so on[3]". Use of 'so on' should not be done here. Names of the important separation methods and their inferiority compared to adsorption with respect to specific application may be added here.
4. Page 3 Line 37: "Experimental materials" Use only "materials" instead of " Experimental materials". The grade/quality, brand/make/purchasing company of all the used chemicals must be mentioned. Authors can refer any paper given in references.
5. Page 4 Line 27: Use "ArsenazoIII spectroscopic method"
6. Page 4 Line 37-38: "respectively" should come after concentrations of U(VI).
7. "Figure. 1A" instead of "figure. 1A" should be used. This may be corrected throughout the manuscript.
8. Page 7 Line 35: How aliquot of kinetics experiments were taken not mentioned either in experimental part or here? Please refer to Journal of Colloid and Interface Science 484 (2016) 196-204 or other paper.
9. Page 8 Line 26: The parameters and variables in the equation are not defined properly.
10. Page 8 Line 52: Correct the caption. Fig. 5A is not "Linear fiiting ..." Write complete caption like "Linear fitting of adsorption isotherms model Langmuir (B) and Freundlich (C) of U(VI) adsorption by BM-SA".
11. Page 9 Line 28: How "Adsorption amount of BM-SA was 0.2 g/L"?
12. Page 12 Line 40-41: Rephrase the sentence. Elaborate the meaning of "good adsorption effect and a certain prospect".

Review form: Reviewer 2

Is the manuscript scientifically sound in its present form?

Yes

Are the interpretations and conclusions justified by the results?

No

Is the language acceptable?

Yes

Do you have any ethical concerns with this paper?

No

Have you any concerns about statistical analyses in this paper?

No

Recommendation?

Major revision is needed (please make suggestions in comments)

Comments to the Author(s)

1. In summary, FTIR is mentioned twice. No details about SEM was given.
2. In the Introduction part, last paragraph 3rd line it is mentioned as "little environmental harm". It can be refined.
3. In the desorption process, adsorbent after getting reacted with HCl, U (VI) is obtained as supernatant. How the U (VI) supernatant will be disposed of? It may require further treatment. It would be helpful if this question is answered?
4. No proper discussion on results was given.
5. For each and every parameter optimization, no proper justification was given for the values obtained such as effect of pH, effect of adsorbent dosage etc.
6. The symbol for pH was not mentioned correctly.
7. Freundlich spelling was wrong, it was mentioned as Friedlich
8. In the fig 4 C, no clear explanation was given for the negative trend in the graph.
9. Graphs should be very clear. The values are not legible.

Decision letter (RSOS-202098.R0)

This year has been very difficult for everyone, and we want to take the opportunity to thank you for your continued support in 2020.

The Royal Society Open Science editorial office will be closed from the evening of Friday 18 December 2020 until Monday 4 January 2021. We will not be responding during this time. If you have received a deadline within this time period, please contact us as soon as possible to allow us to extend the deadline. If you receive any automated messages during this time asking you to meet a deadline, we offer apologies and invite you to respond after the festive period or during normal working hours.

With our best for a peaceful festive period and New Year, and we look forward to working with you in 2021.

Dear Dr Li:

Title: U(VI) adsorption in water by sodium alginate modified *Bacillus megaterium*
Manuscript ID: RSOS-202098

The editor assigned to your manuscript has now received comments from reviewers. We would like you to revise your paper in accordance with the referee and Subject Editor suggestions which can be found below (not including confidential reports to the Editor). Please note this decision does not guarantee eventual acceptance.

Please submit your revised paper before 15-Jan-2021. Please note that the revision deadline will expire at 00.00am on this date. If we do not hear from you within this time then it will be

assumed that the paper has been withdrawn. In exceptional circumstances, extensions may be possible if agreed with the Editorial Office in advance. We do not allow multiple rounds of revision so we urge you to make every effort to fully address all of the comments at this stage. If deemed necessary by the Editors, your manuscript will be sent back to one or more of the original reviewers for assessment. If the original reviewers are not available we may invite new reviewers.

RSC Associate Editor:
Comments to the Author:
(There are no comments.)

RSC Subject Editor:
Comments to the Author:
(There are no comments.)

Reviewers' Comments to Author:
Reviewer: 1

Comments to the Author(s)
In this manuscript, the authors modified *Bacillus megaterium* with sodium alginate, and characterized the material in detail. The sorption of U(VI) was studied under different

experimental conditions and discussed the results in detail. The contents are important for the removal of U(VI) from aqueous solutions. After reading the manuscript, I think it can be accepted for publication after revision.

Special comments:

1. The study is been done only using pure Uranium solution, but no real or simulated solution was used as a case study. The author should use real or simulated solution uranium waste water.
2. In actual waste solution what are the interfering metal ions present and what is the effect of those metal ions on adsorption should also be included.
3. Page 2 Line 56-59: Please correct the sentence, "In recent years, the research on remediation of uranium wastewater pollution has emerged in an endless stream. Mainly contain chemical precipitation, bioremediation (including phytoremediation and microbial remediation), adsorption, and so on[3]". Use of 'so on' should not be done here. Names of the important separation methods and their inferiority compared to adsorption with respect to specific application may be added here.
4. Page 3 Line 37: "Experimental materials" Use only "materials" instead of " Experimental materials". The grade/quality, brand/make/purchasing company of all the used chemicals must be mentioned. Authors can refer any paper given in references.
5. Page 4 Line 27: Use "ArsenazoIII spectroscopic method"
6. Page 4 Line 37-38: "respectively" should come after concentrations of U(VI).
7. "Figure. 1A" instead of "figure. 1A" should be used. This may be corrected throughout the manuscript.
8. Page 7 Line 35: How aliquot of kinetics experiments were taken not mentioned either in experimental part or here? Please refer to Journal of Colloid and Interface Science 484 (2016) 196-204 or other paper.
9. Page 8 Line 26: The parameters and variables in the equation are not defined properly.
10. Page 8 Line 52: Correct the caption. Fig. 5A is not "Linear fitting ..." Write complete caption like "Linear fitting of adsorption isotherms model Langmuir (B) and Freundlich (C) of U(VI) adsorption by BM-SA".
11. Page 9 Line 28: How "Adsorption amount of BM-SA was 0.2 g/L"?
12. Page 12 Line 40-41: Rephrase the sentence. Elaborate the meaning of "good adsorption effect and a certain prospect".

Reviewer: 2

Comments to the Author(s)

1. In summary, FTIR is mentioned twice. No details about SEM was given.
2. In the Introduction part, last paragraph 3rd line it is mentioned as "little environmental harm". It can be refined.
3. In the desorption process, adsorbent after getting reacted with HCl, U (VI) is obtained as supernatant. How the U (VI) supernatant will be disposed of? It may require further treatment. It would be helpful if this question is answered?
4. No proper discussion on results was given.
5. For each and every parameter optimization, no proper justification was given for the values obtained such as effect of pH, effect of adsorbent dosage etc.
6. The symbol for pH was not mentioned correctly.
7. Freundlich spelling was wrong, it was mentioned as Friedlich
8. In the fig 4 C, no clear explanation was given for the negative trend in the graph.
9. Graphs should be very clear. The values are not legible.

Author's Response to Decision Letter for (RSOS-202098.R0)

See Appendix A.

RSOS-202098.R1 (Revision)

Review form: Reviewer 1

Is the manuscript scientifically sound in its present form?

Yes

Are the interpretations and conclusions justified by the results?

Yes

Is the language acceptable?

Yes

Do you have any ethical concerns with this paper?

No

Have you any concerns about statistical analyses in this paper?

No

Recommendation?

Accept as is

Comments to the Author(s)

The authors revised the manuscript carefully and I recommend for publication.

Review form: Reviewer 2

Is the manuscript scientifically sound in its present form?

Yes

Are the interpretations and conclusions justified by the results?

Yes

Is the language acceptable?

Yes

Do you have any ethical concerns with this paper?

No

Have you any concerns about statistical analyses in this paper?

No

Recommendation?

Accept as is

Comments to the Author(s)

Accepted.

Decision letter (RSOS-202098.R1)

Dear Dr Li:

Title: U(VI) adsorption in water by sodium alginate modified *Bacillus megaterium*

Manuscript ID: RSOS-202098.R1

It is a pleasure to accept your manuscript in its current form for publication in Royal Society Open Science. The chemistry content of Royal Society Open Science is published in collaboration with the Royal Society of Chemistry.

RSC Associate Editor:
Comments to the Author:
(There are no comments.)

RSC Subject Editor:
Comments to the Author:
(There are no comments.)

Reviewer(s)' Comments to Author:

Reviewer: 1

Comments to the Author(s)

The authors revised the manuscript carefully and I recommend for publication.

Reviewer: 2

Comments to the Author(s)

Accepted.

Appendix A

Responses to editor and reviewers

(ID: RSOS-202098.R1)

Dear editors and reviewers,

Thank you for your comments concerning our manuscript entitled “U(VI) adsorption in water by sodium alginate modified *Bacillus megaterium*” (ID: RSOS-202098.R1).

We have checked the manuscript carefully and revised it according to the comments. The revised portions are marked in red in the revised manuscript. Here, we resubmit the responses to editors and reviewers.

If you have any questions about the revised manuscript, please don't hesitate to contact me.

Dianxin Li

PhD and Associate Professor

School of Mines and Civil Engineering

Liupanshui Normal University

Liupanshui, Guizhou

P R China 553004

To reviewer #1:

Question 1: The study is been done only using pure Uranium solution, but no real or simulated solution was used as a case study. The author should use real or simulated solution uranium waste water.

Response: The comments from the reviewers are very good. We have supplemented the experiment of U(VI) adsorption by BM-SA in simulated uranium waste water. The geochemical properties of uranium wastewater refer to our previous research

“Journal of Radioanalytical and Nuclear Chemistry 307.2 (2016): 1011-1019” .

Action: Please see “4.8 Uranium removal from simulated uranium wastewater” at the red mark.

Question 2: In actual waste solution what are the interfering metal ions present and what is the effect of those metal ions on adsorption should also be included

Response: The comments from the reviewers are very good. We have supplemented the related research contents of the influence of coexisting ions in the simulated groundwater adsorption experiment.

Action: Please see “4.9 Effect of other interfering ions” at the red mark.

Question 3: Page 2 Line 56-59: Please correct the sentence, "In recent years, the research on remediation of uranium wastewater pollution has emerged in an endless stream. Mainly contain chemical precipitation, bioremediation (including phytoremediation and microbial remediation), adsorption, and so on[3]". Use of 'so on' should not be done here. Names of the important separation methods and their inferiority compared to adsorption with respect to specific application may be added here.

Response: The comments from the reviewers are very good. We have revised the original statement and added relevant contents. The sentence has been changed to “ In recent years, the main remediation methods of uranium wastewater include chemical precipitation, biological remediation (including phytoremediation and microbial remediation), adsorption, and ion exchange. Currently, chemical precipitation method is a usually used method at present, but when the dosage of chemical reagent is excessive, it will cause secondary pollution[3]. Uranium bioremediation in groundwater by microorganism shows that once the electron donors are stopped in the injection well, the reduced U(IV) will be oxidized to U(VI) quickly. It can not achieve the long-term remediation[4]. Phytoremediation is an effective method to treat uranium contamination in topsoil[5]. Adsorption method has the advantages of wide selection of adsorbents, high adsorption efficiency, high

recovery and good economic effect, which has been widely studied by domestic and foreign scholars[6]. ”

Action: Please see the above sentence in “2. Introduction” at the red mark.

Question 4: Page 3 Line 37: “ Experimental materials” Use only "materials" instead of " Experimental materials". The grade/quality, brand/make/purchasing company of all the used chemicals must be mentioned. Authors can refer any paper given in references.

Response: We have used only "Materials" instead of " Experimental materials", and the grade/quality, brand/make/purchasing company of all the used chemicals have been mentioned.

Action: Please see “3.1 Materials” at the red mark.

Question 5: Page 4 Line 27: Use “ ArsenazoIII spectroscopic method ”

Response: We have changed “ Arsenazo III spectrophotometry ” to “ ArsenazoIII spectroscopic method ” .

Action: Please see “3.4 Batch adsorption experiments” at the red mark.

Question 6: Page 4 Line 37-38: “respectively” should come after concentrations of U(VI).

Response: We have changed the sentence to “Where C_0 and C_e were the initial and equilibrium concentrations of U(VI), respectively. V was the volume of U(VI) solution (L), and W was the weight of adsorbent (g).”

Action: Please see “3.4 Batch adsorption experiments” at the red mark.

Question 7: “Figure. 1A ” instead of “ figure. 1A ” should be used. This may be corrected throughout the manuscript.

Response: We have corrected all the similar mistakes in the article.

Action: Please see all the “Figure” at the red mark.

Question 8: Page 7 Line 35: How aliquot of kinetics experiments were taken not mentioned either in experimental part or here? Please refer to Journal of Colloid and Interface Science 484 (2016) 196-204 or other paper.

Response: We have supplemented the experimental conditions and cited the reference. "In the kinetic adsorption experiment, 8 mg SA-GO was added to a 40 ml solution of 50 mg L⁻¹ at pH 5, 30 °C[31]."

Action: Please see "4.3 Effect of reaction time and kinetics" at the red mark.

Question 9: Page 8 Line 26: The parameters and variables in the equation are not defined properly.

Response: We have corrected the parameters and variables.

Action: Please see the "(mg/g)/(L/mg)^{1/n}" in "4.4 Effect of initial U(VI) concentration and adsorption isotherm" at the red mark.

Question 10: Page 8 Line 52: Correct the caption. Fig. 5A is not "Linear fitting ..." Write complete caption like "Linear fitting of adsorption isotherms model Langmuir (B) and Freundlich (C) of U(VI) adsorption by BM-SA".

Response: The comments from the reviewers are very good. We have changed the sentence to "Figure. 5 Adsorption isotherms (A), and linear fitting of isotherms model Langmuir (B) and Frundlich (C) of U(VI) adsorption by BM-SA".

Action: Please see "Figure. 5 Adsorption isotherms (A), and linear fitting of isotherms model Langmuir (B) and Frundlich (C) of U(VI) adsorption by BM-SA" at the red mark.

Question 11: Page 9 Line 28: How "Adsorption amount of BM-SA was 0.2 g/L"?

Response: The most suitable dosage of BM-SA has been calculated as follows: 8 mg (SA-GO) /40 mL(solution)=0.2 g/L. We have added it to the paragraph.

Action: Please see "4.5 Effect of adsorbent dosage" at the red mark.

Question 12: Page 12 Line 40-41: Rephrase the sentence. Elaborate the meaning of

"good adsorption effect and a certain prospect".

Response: The comments from the reviewers are very good. We have changed the sentence to "In conclusion, BM-SA has the advantages of easy synthesis, high removal efficiency, economy and a certain prospect, which was a kind of U(VI) biosorption material for removing low concentration U(VI) in aqueous solution."

Action: Please see "5. Conclusion" at the red mark.

To reviewer #2:

Question 1: In summary, FTIR is mentioned twice. No details about SEM was given.

Response: We have corrected this mistake.

Action: Please see "SEM" in "1.Summary" at the red mark.

Question 2: In the Introduction part, last paragraph 3rd line it is mentioned as "little environmental harm" . It can be refined.

Response: The comments from the reviewers are very good. We have added the sentence "Due to *Bacillus megaterium* is a microorganism existing in the environment, it causes little secondary pollution to the environment."

Action: Please see the above in the last paragraph of "2. Introduction" at the red mark.

Question 3: In the desorption process, adsorbent after getting reacted with HCl, U(VI) is obtained as supernatant. How the U(VI) supernatant will be disposed of? It may require further treatment. It would be helpful if this question is answered?

Response: The concentration of U(VI) in the supernatant was detected to calculate the resolution rate of BM-SA. We added the corresponding description.

Action: Please see "to calculate the desorption percentage of U(VI) adsorbed by

BM-SA.” in “3.5 Desorption and regeneration experiments” at the red mark.

Question 4: No proper discussion on results was given.

Response: We have added discussion in “4. Results and Discussion”.

Action: Please see all the red mark of “4. Results and Discussion”.

Question 5: For each and every parameter optimization, no proper justification was given for the values obtained such as effect of pH, effect of adsorbent dosage etc.

Response: The comments from the reviewers are very good. We have added each and every parameter optimization at Figure. 3~Figure. 6.

Action: Please see the figure legends of Figure. 3~Figure. 6 at the red mark.

Question 6: The symbol for pH was not mentioned correctly.

Response: pH is the strength of acids and alkalis in this paper. We have mentioned it in the last paragraph of “2. Introduction”.

Action: Please see “(the strength of acids and alkalis)” in “3.5 Desorption and regeneration experiments” at the red mark.

Question 7: Friedlich spelling was wrong, it was mentioned as Freundlich

Response: We have changed all the “Friedlich” to “Freundlich” in this paper.

Action: Please see all the “Freundlich” at the red mark in this paper.

Question 8: In the fig 4 C, no clear explanation was given for the negative trend in the graph.

Response: The comments from the reviewers are very good. We have added the explanation for the negative trend in the graph.

Action: Please see “Under certain conditions, q_e was a constant, and q_t increased gradually with time and approached to q_e . So $\ln(q_e - q_t)$ was a decreasing function(Figure. 4C).” in “4.3 Effect of reaction time and kinetics” at the red mark.

Question 9: Graphs should be very clear. The values are not legible.

Response: The comments from the reviewers are very good. We have redrawn the unclear graphs.

Action: Please see Figure. 3 and Figure. 6.